# FreeAnchor: Learning to Match Anchors for Visual Object Detection

Xiaosong Zhang[1], Fang Wan[1], Chang Liu[1], Rongrong Ji[2], Qixiang Ye[1,3*]

[1]University of Chinese Academy of Sciences, Beijing, China
[2]Xiamen University, Xiamen, China    [3]Peng Cheng Laboratory, Shenzhen, China
zhangxiaosong18@mails.ucas.ac.cn, qxye@ucas.ac.cn

## Abstract

Modern CNN-based object detectors assign anchors for ground-truth objects under the restriction of object-anchor Intersection-over-Unit (IoU). In this study, we propose a learning-to-match approach to break IoU restriction, allowing objects to match anchors in a flexible manner. Our approach, referred to as FreeAnchor, updates hand-crafted anchor assignment to "free" anchor matching by formulating detector training as a maximum likelihood estimation (MLE) procedure. FreeAnchor targets at learning features which best explain a class of objects in terms of both classification and localization. FreeAnchor is implemented by optimizing detection customized likelihood and can be fused with CNN-based detectors in a plug-and-play manner. Experiments on COCO demonstrate that FreeAnchor consistently outperforms the counterparts with significant margins[1].

## 1 Introduction

Over the past few years we have witnessed the success of convolution neural network (CNN) for visual object detection [1, 2, 3, 4, 5, 6, 7]. To represent objects with various appearance, aspect ratios, and spatial layouts with limited convolution features, most CNN-based detectors leverage anchor boxes at multiple scales and aspect ratios as reference points for object localization [3, 4, 5, 6, 7]. By assigning each object to a single or multiple anchors, features can be determined and two fundamental procedures, classification and localization (*i.e.*, bounding box regression), are carried out.

Anchor-based detectors leverage spatial alignment, *i.e.*, Intersection over Unit (IoU) between objects and anchors, as the criterion for anchor assignment. Each assigned anchor independently supervises network learning for object prediction, based upon the intuition that the anchors aligned with object bounding boxes are most appropriate for object classification and localization. However, we argue that such intuition is implausible and the hand-crafted IoU criterion is not the best choice.

On the one hand, for objects of acentric features, *e.g.*, slender objects, the most representative features are not close to object centers. A spatially aligned anchor might correspond to fewer representative features, which deteriorate classification and localization capabilities. On the other hand, it is infeasible to match proper anchors/features for objects using IoU when multiple objects come together.

It is hard to design a generic rule which can optimally match anchors/features with objects of various geometric layouts. The widely used hand-crafted assignment could fail when facing acentric, slender, and/or crowded objects. A learning-based approach requires to be explored to solve this problem in a systematic way, which is the focus of this study.

We propose a learning-to-match approach for object detection, and target at discarding hand-crafted anchor assignment while optimizing learning procedures of visual object detection from three specific aspects. First, to achieve a high recall rate, the detector is required to guarantee that for each object at least one anchor's prediction is close to the ground-truth. Second, in order to achieve high detection precision, the detector needs to classify anchors with poor localization (large bounding box regression error) into background. Third, the predictions of anchors should be compatible with the non-maximum suppression (NMS) procedure, $i.e.$, the higher the classification score is, the more accurate the localization is. Otherwise, an anchor with accurate localization but low classification score could be suppressed when using the NMS process.

To fulfill these objectives, we formulate object-anchor matching as a maximum likelihood estimation (MLE) procedure [8, 9], which selects the most representative anchor from a "bag" of anchors for each object. We define the likelihood probability of each anchor bag as the largest anchor confidence within it. Maximizing the likelihood probability guarantees that there exists at least one anchor, which has high confidence for both object classification and localization. Meanwhile, most anchors, which have large classification or localization error, are classified as background. During training, the likelihood probability is converted into a loss function, which then drives CNN-based detector training and object-anchor matching.

The contributions of this work are concluded as follows:

- **We formulate detector training as an MLE procedure and update hand-crafted anchor assignment to free anchor matching.** The proposed approach breaks the IoU restriction, allowing objects to flexibly select anchors under the principle of maximum likelihood.

- **We define a detection customized likelihood, and implement joint optimization of object classification and localization in an end-to-end mechanism.** Maximizing the likelihood drives network learning to match optimal anchors and guarantees the comparability of with the NMS procedure.

## 2   Related Work

Object detection requires generating a set of bounding boxes along with their classification labels associated with objects in an image. However, it is not trivial for a CNN-based detector to directly predict an order-less set of arbitrary cardinals. One widely-used workaround is to introduce anchors, which employs a divide-and-conquer process to match objects with features. This approach has been successfully demonstrated in Faster R-CNN [3], SSD [5], FPN [6], RetinaNet [7], DSSD [10] and YOLOv2 [11]. In these detectors, dense anchors need to be configured over convolutional feature maps so that features extracted from anchors can match object windows and the bounding box regression can be well initialized. Anchors are then assigned to objects or backgrounds by thresholding their IoUs with ground-truth bounding boxes [3].

Although effective, these approaches are restricted by heuristics that spatially aligned anchors are compatible for both object classification and localization. For objects of acentric features, however, the detector could miss the best anchors and features.

To break this limitation imposed by pre-assigned anchors, recent anchor-free approaches employ pixel-level supervision [12] and center-ness bounding box regression [13]. CornerNet [14] and CenterNet [15] replace bounding box supervision with key-point supervision. MetaAnchor [16] approach learns to produce anchors from the arbitrary customized prior boxes with a sub-network. GuidedAnchoring [17] leverages semantic features to guide the prediction of anchors while replacing dense anchors with predicted anchors. IoU-Net [18] incorporates IoU-guided NMS, which helps eliminating the suppression failure caused by the misleading classification confidences.

Existing approaches have taken a step towards learnable anchor customization. Nevertheless, to the best of our knowledge, there still lacks a systematic approach to model the correspondence between anchors and objects during detector training, which inhibits the optimization of feature selection and feature learning.

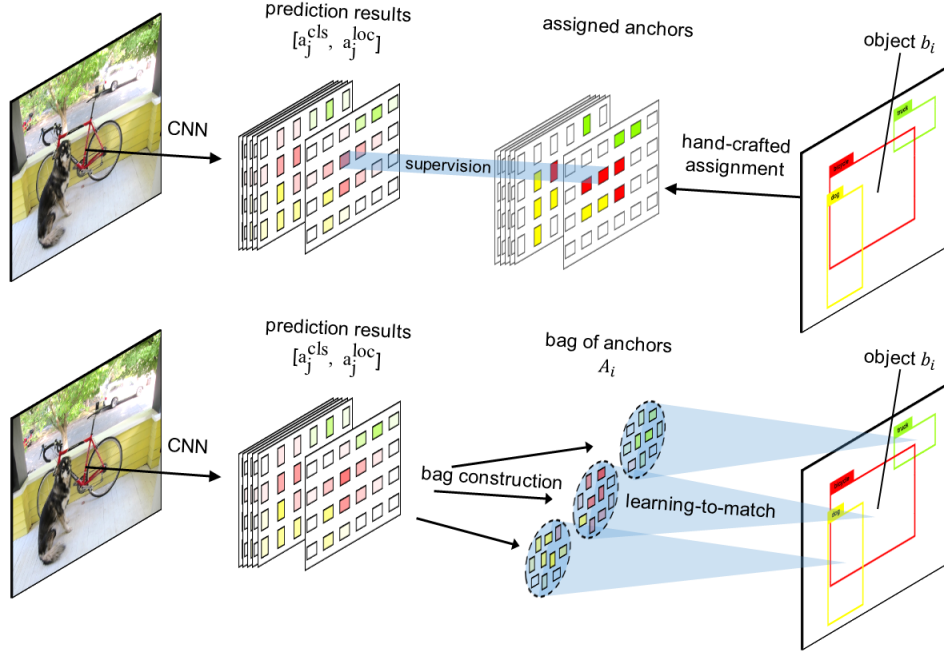

Figure 1: Comparison of hand-crafted anchor assignment (top) and FreeAnchor (bottom). FreeAnchor allows each object to flexibly match the best anchor from a "bag" of anchors during detector training.

## 3 The Proposed Approach

To model the correspondence between objects and anchors, we propose to formulate detector training as an MLE procedure. We then define the detection customized likelihood, which simultaneously facilitates object classification and localization. During detector training, we convert detection customized likelihood into detection customized loss and jointly optimizing object classification, object localization, and object-anchor matching in an end-to-end mechanism.

### 3.1 Detector Training as Maximum Likelihood Estimation

Let's begin with a CNN-based one-stage detector [7]. Given an input image $I$, the ground-truth annotations are denoted as $B$, where a ground-truth box $b_i \in B$ is made up of a class label $b_i^{cls}$ and a location $b_i^{loc}$. During the forward propagation procedure of the network, each anchor $a_j \in A$ obtains a class prediction $a_j^{cls} \in \mathcal{R}^k$ after the Sigmoid activation, and a location prediction $a_j^{loc} = \{x, y, w, h\}$ after the bounding box regression. $k$ denotes the number of object classes.

During training, hand-crafted criterion based on IoU is used to assign anchors for objects, Fig. 1, and a matrix $C_{ij} \in \{0, 1\}$ is defined to indicate whether object $b_i$ matches anchor $a_j$ or not. When the IoU of $b_i$ and $a_j$ is greater than a threshold, $b_i$ matches $a_j$ and $C_{ij} = 1$. Otherwise, $C_{ij} = 0$. Specially, when multiple objects' IoU are greater than this threshold, the object of the largest IoU will successfully match this anchor, which guarantees that each anchor is matched by a single object at most, $i.e.$, $\sum_i C_{ij} \in \{0, 1\}, \forall a_j \in A$. By defining $A_+ \subseteq A$ as $\{a_j \mid \sum_i C_{ij} = 1\}$ and $A_- \subseteq A$ as $\{a_j \mid \sum_i C_{ij} = 0\}$, the loss function $\mathcal{L}(\theta)$ of the detector is written as follows:

$$\mathcal{L}(\theta) = \sum_{a_j \in A_+} \sum_{b_i \in B} C_{ij} \mathcal{L}_{ij}^{cls}(\theta) + \beta \sum_{a_j \in A_+} \sum_{b_i \in B} C_{ij} \mathcal{L}_{ij}^{loc}(\theta) + \sum_{a_j \in A_-} \mathcal{L}_j^{bg}(\theta), \qquad (1)$$

where $\theta$ denotes the network parameters to be learned. $\mathcal{L}_{ij}^{cls}(\theta) = BCE(a_j^{cls}, b_i^{cls}, \theta)$, $\mathcal{L}_{ij}^{loc}(\theta) = SmoothL1(a_j^{loc}, b_i^{loc}, \theta)$ and $\mathcal{L}_j^{bg}(\theta) = BCE(a_j^{cls}, \vec{0}, \theta)$ respectively denote the Binary Cross Entropy loss ($BCE$) for classification and the $SmoothL1$ loss defined for localization [2]. $\beta$ is a regularization factor and "bg" indicates "background".

From the MLE perspective, the training loss $\mathcal{L}(\theta)$ is converted into a likelihood probability, as follows:

$$
\begin{aligned}
\mathcal{P}(\theta) &= e^{-\mathcal{L}(\theta)} \\
&= \prod_{a_j \in A_+} \big( \sum_{b_i \in B} C_{ij} e^{-\mathcal{L}_{ij}^{cls}(\theta)} \big) \prod_{a_j \in A_+} \big( \sum_{b_i \in B} C_{ij} e^{-\beta \mathcal{L}_{ij}^{loc}(\theta)} \big) \prod_{a_j \in A_-} e^{-\mathcal{L}_j^{bg}(\theta)} \\
&= \prod_{a_j \in A_+} \big( \sum_{b_i \in B} C_{ij} \mathcal{P}_{ij}^{cls}(\theta) \big) \prod_{a_j \in A_+} \big( \sum_{b_i \in B} C_{ij} \mathcal{P}_{ij}^{loc}(\theta) \big) \prod_{a_j \in A_-} \mathcal{P}_j^{bg}(\theta),
\end{aligned}
\tag{2}
$$

where $\mathcal{P}_{ij}^{cls}(\theta)$ and $\mathcal{P}_j^{bg}(\theta)$ denote classification confidence and $\mathcal{P}_{ij}^{loc}(\theta)$ denotes localization confidence. Minimizing the loss function $\mathcal{L}(\theta)$ defined in Eq. 1 is equal to maximizing the likelihood probability $\mathcal{P}(\theta)$ defined in Eq. 2.

Eq. 2 strictly considers the optimization of classification and localization of anchors from the MLE perspective. However, it unfortunately ignores how to learn the matching matrix $C_{ij}$. Existing CNN-based detectors [3, 5, 6, 7, 11] solve this problem by empirically assigning anchors using the IoU criterion, Fig. 1, but ignoring the optimization of object-anchor matching.

## 3.2 Detection Customized Likelihood

To achieve the optimization of object-anchor matching, we extend the CNN-based detection framework by introducing detection customized likelihood. Such likelihood intends to incorporate the objectives of recall and precision while guaranteeing the compatibility with NMS.

To implement the likelihood, we first construct a bag of candidate anchors for each object $b_i$ by selecting $(n)$ top-ranked anchors $A_i \subset A$ in terms of their IoU with the object. We then learns to match the best anchor while maximizing the detection customized likelihood.

To optimize the recall rate, for each object $b_i \in B$ we requires to guarantee that there exists at least one anchor $a_j \in A_i$, whose prediction ($a_j^{cls}$ and $a_j^{loc}$) is close to the ground-truth. The objective function can be derived from the first two terms of Eq. 2, as follows:

$$
\mathcal{P}_{recall}(\theta) = \prod_i \max_{a_j \in A_i} \big( \mathcal{P}_{ij}^{cls}(\theta) \mathcal{P}_{ij}^{loc}(\theta) \big).
\tag{3}
$$

To achieve increased detection precision, detectors need to classify the anchors of poor localization into the background class. This is fulfilled by optimizing the following objective function:

$$
\mathcal{P}_{precision}(\theta) = \prod_j \big( 1 - P\{a_j \in A_-\}(1 - \mathcal{P}_j^{bg}(\theta)) \big),
\tag{4}
$$

where $P\{a_j \in A_-\} = 1 - \max_i P\{a_j \to b_i\}$ is the probability that $a_j$ misses all objects and $P\{a_j \to b_i\}$ denotes the probability that anchor $a_j$ correctly predicts object $b_i$.

To be compatible with the NMS procedure, $P\{a_j \to b_i\}$ should have the following three properties: (1) $P\{a_j \to b_i\}$ is a monotonically increasing function of the IoU between $a_j^{loc}$ and $b_i$, $IoU_{ij}^{loc}$. (2) When $IoU_{ij}^{loc}$ is smaller than a threshold $t$, $P\{a_j \to b_i\}$ is close to 0. (3) For an object $b_i$, there exists one and only one $a_j$ satisfying $P\{a_j \to b_i\} = 1$. These properties can be satisfied with a saturated linear function, as

$$
\text{Saturated linear}(x, t_1, t_2) = \begin{cases} 0, & x \le t_1 \\ \dfrac{x - t_1}{t_2 - t_1}, & t_1 < x < t_2, \\ 1, & x \ge t_2 \end{cases}
$$

which is shown in Fig. 2, and we have $P\{a_j \to b_i\} = \text{Saturated linear}\big(IoU_{ij}^{loc}, t, \max_j(IoU_{ij}^{loc})\big)$.

Implementing the definitions provided above, the detection customized likelihood is defined as follows:

$$
\begin{aligned}
\mathcal{P}'(\theta) &= \mathcal{P}_{recall}(\theta) \times \mathcal{P}_{precision}(\theta) \\
&= \prod_i \max_{a_j \in A_i} (\mathcal{P}_{ij}^{cls}(\theta) \mathcal{P}_{ij}^{loc}(\theta)) \times \prod_j \big( 1 - P\{a_j \in A_-\}(1 - \mathcal{P}_j^{bg}(\theta)) \big),
\end{aligned}
\tag{5}
$$

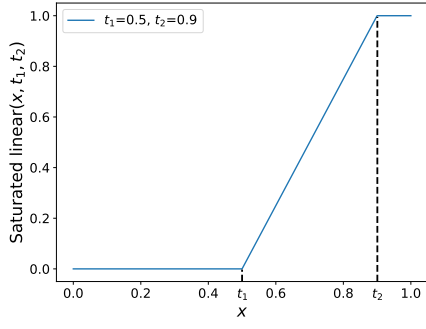

Figure 2: Saturated linear function.

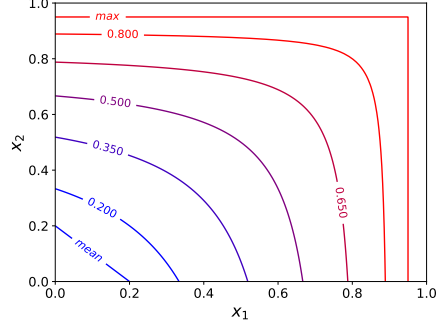

Figure 3: Mean-max function.

which incorporates the objectives of recall, precision and compatibility with NMS. By optimizing this likelihood, we simultaneously maximize the probability of recall $\mathcal{P}_{recall}(\theta)$ and precision $\mathcal{P}_{precision}(\theta)$ and then achieve free object-anchor matching during detector training.

### 3.3 Anchor Matching Mechanism

To implement this learning-to-match approach in a CNN-based detector, the detection customized likelihood defined by Eq. 5 is converted to a detection customized loss function, as follows:

$$
\begin{aligned}
\mathcal{L}'(\theta) &= -\log \mathcal{P}'(\theta) \\
&= -\sum_i \log \big( \max_{a_j \in A_i} (\mathcal{P}_{ij}^{cls}(\theta) \mathcal{P}_{ij}^{loc}(\theta)) \big) - \sum_j \log \big( 1 - P\{a_j \in A_-\}(1 - \mathcal{P}_j^{bg}(\theta)) \big),
\end{aligned} \quad (6)
$$

where the max function is used to select the best anchor for each object. During training, a single anchor is selected from a bag of anchors $A_i$, which is then used to update the network parameter $\theta$.

At early training epochs, the confidence of all anchors is small for randomly initialized network parameters. The anchor with the highest confidence is not suitable for detector training. We therefore propose using the Mean-max function, defined as:

$$
\text{Mean-max}(X) = \frac{\sum_{x_j \in X} \dfrac{x_j}{1 - x_j}}{\sum_{x_j \in X} \dfrac{1}{1 - x_j}},
$$

which is used to select anchors. When training is insufficient, the Mean-max function, as shown in Fig. 3, will be close to the mean function, which means almost all anchors in bag are used for training. Along with training, the confidence of some anchors increases and the Mean-max function moves closer to the max function. When sufficient training has taken place, a single best anchor can be selected from a bag of anchors to match each object.

Replacing the max function in Eq. 6 with Mean-max, adding balance factor $w_1$ $w_2$, and applying focal loss [7] to the second term of Eq. 6, the detection customized loss function of an FreeAnchor detector is concluded, as follows:

$$
\mathcal{L}''(\theta) = -w_1 \sum_i \log \big( \text{Mean-max}(X_i) \big) + w_2 \sum_j FL \big( P\{a_j \in A_-\}(1 - \mathcal{P}_j^{bg}(\theta)) \big), \quad (7)
$$

where $X_i = \{\mathcal{P}_{ij}^{cls}(\theta)\mathcal{P}_{ij}^{loc}(\theta)| \ a_j \in A_i\}$ is a likelihood set corresponding to the anchor bag $A_i$. By using the parameters $\alpha$ and $\gamma$ from focal loss [7], we set $w_1 = \frac{\alpha}{||B||}$ , $w_2 = \frac{1-\alpha}{n||B||}$, and $FL(x) = -x^\gamma \log(1-x)$.

With the detection customized loss defined, we implement the training procedure as Algorithm 1.

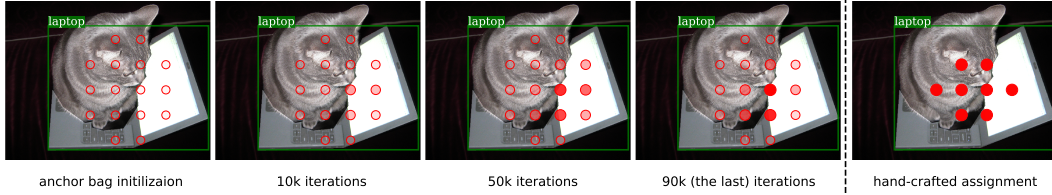

| anchor bag initilizaion | 10k iterations | 50k iterations | 90k (the last) iterations | hand-crafted assignment |

Figure 4: Comparison of learning-to-match anchors (left) with hand-crafted anchor assignment (right) for the "laptop" object. Red dots denote anchor centers. Darker (redder) dots denote higher confidence to be matched. For clarity, we select 16 anchors of aspect-ratio 1:1 from all 50 anchors for illustration. (Best viewed in color)

---

**Algorithm 1** Detector training with FreeAnchor.

---

**Input:**    $I$: Input image.
         $\mathcal{B}$: A set of ground-truth bounding boxes $b_i$.
         $\mathcal{A}$: A set of anchors $a_j$ in image.
         $n$: Hyper-parameter about anchor bag size .
**Output:**   $\theta$: Detection network parameters.
 1:  $\theta \leftarrow$ initialize network parameters.
 2:  **for** i=1:MaxIter **do**
 3:     **Forward propagation:**
         Predict class $a_j^{cls}$ and location $a_j^{loc}$ for each anchor $a_j \in \mathcal{A}$.
 4:     **Anchor bag construction:**
         $\mathcal{A}_i \leftarrow$ Select $n$ top-ranked anchors $a_j$ in terms of their IoU with $b_i$.
 5:     **Loss calculation:**
         Calculate $L''(\theta)$ with Eq. 7.
 6:     **Backward propagation:**
         $\theta^{t+1} = \theta^t - \lambda \nabla_{\theta^t} L''(\theta^t)$ using a stochastic gradient descent algorithm.
 7:  **end for**
 8:  **return** $\theta$

---

# 4 Experiments

In this section, we present the implementation of an FreeAnchor detector to appraise the effect of the proposed learning-to-match approach. We also compare the FreeAnchor detector with the counterpart and the state-of-the-art approaches. Experiments were carried out on COCO 2017[19], which contains $\sim$118k images for training, 5k for validation (val) and $\sim$20k for testing without provided annotations ($test\text{-}dev$). Detectors were trained on COCO training set, and evaluated on the $val$ set. Final results were reported on the $test\text{-}dev$ set.

## 4.1 Implementation Details

FreeAnchor is implemented upon a state-of-the-art one-stage detector, RetinaNet [7], using ResNet [20] and ResNeXt [21] as the backbone networks. By simply replacing the loss defined in RetinaNet with the proposed detection customized loss, Eq. 7, we updated the RetinaNet detector to an FreeAnchor detector. For the last convolutional layer of the classification subnet, we set the bias initialization to $b = -\log((1-\rho)/\rho)$ with $\rho = 0.02$. Training used synchronized SGD over 8 Tesla V100 GPUs with a total of 16 images per mini-batch (2 images per GPU). Unless otherwise specified, all models were trained for 90k iterations with an initial learning rate of 0.01, which is then divided by 10 at 60k and again at 80k iterations.

## 4.2 Model Effect

**Learning-to-match:** The proposed learning-to-match approach can select proper anchors to represent the object of interest, Fig. 4. As analyzed in the introduction section, hand-crafted anchor assignment often fails in two situations: Firstly, slender objects with acentric features; and secondly when multiple

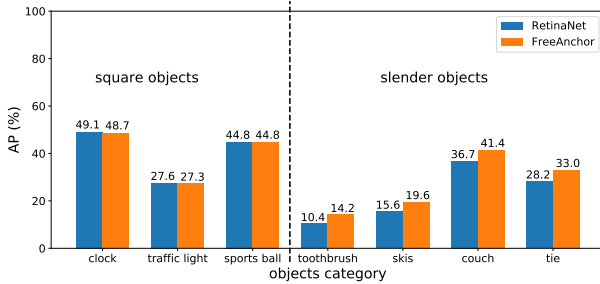 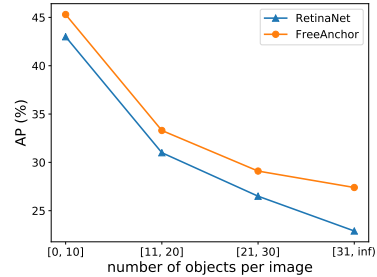

Figure 5: Performance comparison on square and slender objects.

Figure 6: Performance comparison on object crowdedness.

objects are provided in crowded scenes. FreeAnchor effectively alleviated these two problems. Over slender objects, FreeAnchor significantly outperformed the RetinaNet baseline, Fig. 5. For other square objects FreeAnchor reported comparable performance with RetinaNet. The reason for this is that the learning-to-match procedure drives network activating at least one anchor within each object's anchor bag in order to predict correct category and location. The anchor is not necessary spatially aligned with the object, but has the most representative features for object classification and localization.

We further compared the performance of RetinaNet and FreeAnchor in scenarios of various crowdedness, Fig. 6. As the number of objects in each image increased, the FreeAnchor's advantage over RetinaNet became more and more obvious. This demonstrated that our approach, with the learning-to-match mechanism, can select more suitable anchors to objects in crowded scenes.

**Compatibility with NMS:** To assess the compatibility of anchors' predictions with NMS, we defined the NMS Recall ($NR_\tau$) as the ratio of the recall rates after and before NMS for a given IoU thresholds $\tau$. Following the COCO-style AP metric [19], NR was defined as the averaged $NR_\tau$ when $\tau$ changes from 0.50 to 0.90 with an interval of 0.05, Table 1. We compared RetinaNet and FreeAnchor in terms of their $NR_\tau$. It can be seen that FreeAnchor reported higher $NR_\tau$, which means higher compatibility with NMS. This validated that the detection customized likelihood, defined in Section 3.2, can drive joint optimization of classification and localization.

Table 1: Comparison of NMS recall (%) on COCO *val* set.

| backbone | detector | NR | $NR_{50}$ | $NR_{60}$ | $NR_{70}$ | $NR_{80}$ | $NR_{90}$ |
|---|---|---|---|---|---|---|---|
| ResNet-50 | RetinaNet [7] | 81.8 | 98.3 | 95.7 | 87.0 | 71.8 | 51.3 |
| | FreeAnchor (ours) | **83.8** | **99.2** | **97.5** | **89.5** | **74.3** | **53.1** |

## 4.3 Parameter Setting

**Anchor bag size** $n$**:** We evaluated anchor bag sizes in {40, 50, 60, 100} and observed that the bag size 50 reported the best performance.

**Background IoU threshold** $t$**:** A threshold was used in $P\{a_j \rightarrow b_i\}$ during training. We tried background IoU thresholds in {0.5, 0.6, 0.7} and validated that 0.6 worked best.

**Focal loss parameter:** FreeAnchor introduced a bag of anchors to replace independent anchors and therefore faced more serious sample imbalance. To handle the imbalance, we experimented the parameters in Focal Loss [7] as $\alpha$ in {0.25, 0.5, 0.75} and $\gamma$ in {1.5 , 2.0, 2.5}, and set $\alpha = 0.5$ and $\gamma = 2.0$.

**Loss regularization factor** $\beta$**:** The regularization factor $\beta$ in Eq. 1, which balances the loss of classification and localization, was experimentally validated to be 0.75.

## 4.4 Detection Performance

In Table 2, FreeAnchor was compared with the RetinaNet baseline. FreeAnchor consistently improved the AP up to ~3.0%, which is a significant margin in terms of the challenging object detection task. Note that the performance gain was achieved with negligible cost of training time.

Table 2: Performance comparison of FreeAnchor and RetinaNet (baseline).

| Backbone | Detector | Training time | AP | $AP_{50}$ | $AP_{75}$ | $AP_S$ | $AP_M$ | $AP_L$ |
|---|---|---|---|---|---|---|---|---|
| ResNet-50 | RetinaNet [7] | 5.02h | 35.7 | 55.0 | 38.5 | 18.9 | 38.9 | 46.3 |
| | FreeAnchor (ours) | 5.27h | **38.7** | **57.3** | **41.6** | **20.2** | **41.3** | **50.1** |
| ResNet-101 | RetinaNet [7] | 6.96h | 37.8 | 57.5 | 40.8 | 20.2 | 41.1 | 49.2 |
| | FreeAnchor (ours) | 7.26h | **40.9** | **59.9** | **43.8** | **21.7** | **43.8** | **53.0** |

FreeAnchor was compared with state-of-the-art one-stage detectors in Table 3, used scale jitter and 2× longer training than the same model from Table 2. It outperformed the baseline RetinaNet [7] and the anchor-free approaches including FoveaBox [22], FSAF [23], FCOS [13] and CornerNet [14]. With a litter ResNeXt-64x4d-101 backbone network and fewer training iterations, FreeAnchor was comparable with CenterNet in AP (44.9% vs. 44.9%) and reported higher $AP_{50}$, which is a more widely used metric in many applications.

"FreeAnchor*" refers to extending the scale range from [640, 800] to [480, 960], achieving 46.0% AP. "FreeAnchor**" further utilized multi-scale testing over scales {480, 640, 800, 960, 1120, 1280}, and increased AP up to 47.3%, which outperformed most state-of-the-art detectors with the same backbone network.

Table 3: Performance comparison with state-of-the-art one-stage detectors.

| Detector | Backbone | Iter. | AP | $AP_{50}$ | $AP_{75}$ | $AP_S$ | $AP_M$ | $AP_L$ |
|---|---|---|---|---|---|---|---|---|
| RetinaNet [7] | ResNet-101 | 135k | 39.1 | 59.1 | 42.3 | 21.8 | 42.7 | 50.2 |
| FoveaBox [22] | ResNet-101 | 135k | 40.6 | 60.1 | 43.5 | 23.3 | 45.2 | 54.5 |
| FSAF [23] | ResNet-101 | 135k | 40.9 | 61.5 | 44.0 | 24.0 | 44.2 | 51.3 |
| FCOS [13] | ResNet-101 | 180k | 41.5 | 60.7 | 45.0 | 24.4 | 44.8 | 51.6 |
| RetinaNet [7] | ResNeXt-101 | 135k | 40.8 | 61.1 | 44.1 | 24.1 | 44.2 | 51.2 |
| FoveaBox [22] | ResNeXt-101 | 135k | 42.1 | 61.9 | 45.2 | 24.9 | 46.8 | 55.6 |
| FSAF [23] | ResNeXt-101 | 135k | 42.9 | 63.8 | 46.3 | 26.6 | 46.2 | 52.7 |
| FCOS [13] | ResNeXt-101 | 180k | 43.2 | 62.8 | 46.6 | 26.5 | 46.2 | 53.3 |
| CornerNet [14] | Hourglass-104 | 500k | 40.6 | 56.4 | 43.2 | 19.1 | 42.8 | 54.3 |
| CenterNet [15] | Hourglass-104 | 480k | 44.9 | 62.4 | 48.1 | 25.6 | 47.4 | 57.4 |
| FreeAnchor | ResNet-101 | 180k | 43.1 | 62.2 | 46.4 | 24.5 | 46.1 | 54.8 |
| FreeAnchor | ResNeXt-101 | 180k | 44.9 | 64.3 | 48.5 | 26.8 | 48.3 | 55.9 |
| FreeAnchor* | ResNeXt-101 | 180k | 46.0 | 65.6 | 49.8 | 27.8 | 49.5 | 57.7 |
| FreeAnchor** | ResNeXt-101 | 180k | **47.3** | **66.3** | **51.5** | **30.6** | **50.4** | **59.0** |

## 5 Conclusion

We proposed an elegant and effective approach, referred to as FreeAnchor, for visual object detection. FreeAnchor updated the hand-crafted anchor assignment to "free" object-anchor correspondence by formulating detector training as a maximum likelihood estimation (MLE) procedure. With FreeAnchor implemented, we significantly improved the performance of object detection, in striking contrast with the baseline detector. The underlying reality is that the MLE procedure with the detection customized likelihood facilitates learning convolutional features that best explain a class of objects. This provides a fresh insight for the visual object detection problem.

**Acnkowledgement.** This work was supported in part by the NSFC under Grant 61836012, 61671427, and 61771447 and Post Doctoral Innovative Talent Support Program under Grant 119103S304.

## Footnotes

[1]Code is available at https://github.com/zhangxiaosong18/FreeAnchor

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
