[Supplementary Material]

# Supplementary Material for FreeAnchor

## 1 The code for FreeAnchor

The files in the directory "Code for FreeAnchor" contain implementation details of our approach, which can be reproduced by following the "README.md" file.

## 2 Detection examples on MS-COCO *val* set

**Left:** RetinaNet-ResNeXt-101 **Right:** FreeAnchor-ResNeXt-101

**Red boxes:** By FreeAnchor **Blue boxes:** By FreeAnchor and RetinaNet

### 2.1 Slender objects

 **2.2 Occluded objects**

 ## 2.3 Crowded objects