[Reviews · NeurIPS 2019]

Reviewer 1



UPDATED post rebuttal: Thanks to the authors for addressing all my points. I am raising my score to seven. The authors begin by noting that many existing object detection pipelines include a step on 'anchor assignment', where from a large set of candidate bounding boxes (or "anchors") in a generic image frame, the one that best matches the ground truth bounding box, as measure by IoU, is chosen to be the one that is used for training, ie the object detection and bounding box regression outputs for that anchor will be pushed towards the ground truth. The authors note that for objects which don't fill the anchor well (slim objects oriented diagonally, objects with holes, or occluded objects) the best anchor according to this IoU comparison may be actively bad for training as a whole. The authors propose "learning to match", ie producing a custom likelihood which promotes both precision and recall of the final result (making reference to terms from the traditional loss function). For each ground truth bounding box, a 'bag of anchors' is selected by ranking IoU and picking the best n. During training, a different bounding box is selected from this bag for each object, for each backwards pass. Which one is chosen depends on the current state of training - as demonstrated in Figure 4, the confidence gradually increases for certain anchors. The Mean-Max function means that at the start of training, many of the anchors in the bag will be selected, but over time a single best one will come to dominate. I do not have the relevant background to confidently assess originality / quality / significance in this subfield. The results seem impressive, to do a drop-in replacement of the loss function and get multiple percent increases on difficult categories (figure 5 right) with negligible impact on other classes is a good result. Figure 6 is a nice result as well. All necessary experimental details seem to be present. The paper seems to be well written, and for those working with these kind of models I'm fairly confident that trying out these changes would be simple, given the information in the paper. Minor points: * L16: the citations given for algorithms which incorporate anchor boxes includes [5] and [6], which are R-CNN and Fast R-CNN - neither of these papers includes the term 'anchor', I believe that first came into that line of work as part of Faster R-CNN. * In the figure 1 caption, (top) and (bottom) would read better to me than (up) and (down). * L145: "The anchor with the highest confidence is not be suitable for detector training" - remove the extra word "be" * Algorithm 1 - the anchor bag construction is implied to happen on every forwards pass - but if it just depends on the IoU between the anchor and the ground truth bounding box, presumably this could be done once before the training loop and cached? * Algorithm 1 - "super-parameter" - I'm pretty sure I have not come across this term before, and google search says "did you mean hyperparameter"... * Algorithm 1 - Backward propagation - I think it's way more standard to do something like $\theta^{t+1} = \theta^t - \lambda \nabla_{\theta^t}L(\theta^t)$. You are presumably not solving the exact argmin problem, and it takes approximately as much space to write. * Figure 4 - Firstly, I would recommend noting in the caption that the laptop is relatively low contrast and encouraging readers to zoom in. I am viewing in color and it was on the 3rd pass through the paper that I realised the green bounding box does actually capture the laptop very well - I basically didn't notice half the laptop and it's easy to assume that this is some kind of failure case where it's incorrectly focused on the cat. Given this realisation, I'm still not sure if this image shows as much as it could - obviously there is progression to the right as the center anchors get redder, but I don't really know what actual spatial extent those anchors represent. It's also unclear why more than these 16 were not represented (the minimum anchor bag size that is mentioned is 40) - presumably the confidence goes really low much further away, but could we then see this? Perhaps it might be more interesting to show, in separate images, the actual anchor extents for the final most confident and least confident. My intuition is that one of them would clearly have a better IoU with the true bounding box, but the eventual higher confidence one would focus more on the pixels that are a laptop - am I right about this? * L172 - I am intrigued as to why this value for a bias initialization (presumably the rest of the convolutional biases are initialized at zero as normal). Can you provide some justification as to why this formula is used? I would also recommend not using $\pi$, as that already has a well known scalar interpretation.

Reviewer 2



Overall, the introduced method is interesting and novel, however, the paper is not yet in a state to be acceptable for publication. Some parts of the paper appear too ad-hoc and it is not not always clear what the contribution of the introduced method is. More details below: - The third and fourth paragraph of Section 1 appear convoluted and are difficult to follow. It should be more clear what the motivation of the introduced technique is. - Section 2: it is not sufficiently explained how the presented approach differs from existing methods, e.g. such as FoveaBox or MetaAnchor. This should clarified. - The text states 'The proposed approach breaks the IoU restriction, allowing objects to flexibly select anchors ...". However, in Section 3, it then states 'a hand-crafted criterion based on IoU is used.' This is confusing. What for was the IoU criterion used? What is similar/different to existing approaches. - a^{loc}_j is defined to be in R4. Which dimensions are defined here? It later becomes clear from the context, but it should be more formally defined. - The definitions in lines 101, 102 are ambiguous and should be clarified. How is SmoothL1 defined? - 'bg' as e.g. in \mathcal(\theta)^{bg} is not defined. - Why do \mathcal{P}(\theta)^{cls}_{ij} \mathcal{P}(\theta)^{bg}_{j} together define class confidence? - Lines 110 - 113: If Eq. 2 strictly considers the optimization of classification and localization of anchors but it ignores how to learn the matching matrix C what is the conclusion here? Is this a limitation? - L121: 'it requires to guarantee...'. What requires to guarantee? - Some equations (e.g. Eq. 4) are not necessary to explain the contribution of the paper and should be moved to the Appendix. This would help to sharpen the presentation of the contribution. Similarly for Eq. 5 and 6. - It is not clear what is the relationship of the Mean-max function and insufficient training. Why will it be close to the Mean when training is not sufficient? What is X? It becomes clear later in the text, but as presented this appears too ad-hoc. - Figure 2 and 3 could be moved to the Appendix to make room for more important figures. - What is meant by 'inheriting the parameters' (L156)? - L200: Is is not clear how a 'bag' of anchors was constructed and how they where selected. - The quantitative evaluation discussed in Section 4.2 (and the qualitative results in Supplementary Material) is interesting. However, it would have been more informative to also provide the average performance across all categories. Also, it is not clear why 'couch' is considered a slender object. - The results presented in Table 3 are interesting but the presented approach only marginally improve upon existing work or not at all. The cases for which it works well and does not are not sufficiently discussed. - No limitations are discussed. In which situations does the approach fail, also compared to existing work?

Reviewer 3



The paper propose a free-anchor mechanism to address the object detection and break the constrain that anchors should be pre-defined and static. The proposed method updates anchor assignment to the dynamic matching by MLE. Overall this Is a nice paper to resolve the anchor assignment based on the Maximum likelihood instead of the alternative IoU criterion, which is widely used in previous methods. The paper is written in a clear manner in terms of organisation and logic. The core of the novelty in the paper is the assignment of anchors - based on MLE rather than the rigid IoU-based criterion. Such a modification is proven to be better achieving the goal of “anchor-object matching”. The traditional IoU matching criterion could result in some confusing assignment especially the objects are heavily occluded. The proposed maximum likelihood solution could somehow alleviate such a situation.

[Author Response · NeurIPS 2019]

# 1 Paper strengths

**R1:**"Performance boosts on many whole-dataset metrics." "FreeAnchor performs notably better than the baseline..." **R2:**"Overall, the introduced method is interesting and novel..." "This paper could have impact if the overall presentation would be improved." **R3:** "Overall this is a nice paper to resolve the anchor assignment based on the Maximum likelihood...".

We thank reviewers for their valuable comments and support of the novelty and results of our work. Major concerns are mainly about presentation and explanation, which are addressed point-to-point below.

# 2 Response to review comments

**R1: Could the anchor bag be constructed once before the training loop and cached?** Yes. Nevertheless, we can call it in each iteration for negligible computational cost and simplicity of implementation.

**R1: Figure 4.** Yes you are right. As suggested, we improved the contrast and explanation of Figure 4, as below.

| anchor bag initilizaion | 10k iterations | 50k iterations | 90k (the last) iterations | hand-crafted assignment |

Figure 4: Comparison of learning-to-match anchors (left) with hand-crafted anchor assignment (right) for the "laptop" object. Red dots denote anchor centers. Darker (redder) dots denote higher confidence to be matched. For clarity, we select 16 anchors of aspect-ratio 1:1 from all 40 anchors for illustration. (Best viewed in color)

**R1: L172, why this formula is used? & why $b$ is used for a bias initialization** According to Focal Loss [12], the formula $(b)$ is the inverse function of Sigmoid. $b$, as a bias initialization, can alleviate the class imbalance issue.

**R1&R3: Minor comments.** L16, [5] and [6] were removed; Fig. 1, "up" $\rightarrow$ "top", "down" $\rightarrow$ "bottom"; L145, "be" is removed; Algorithm 1, "super-" $\rightarrow$ "hyper-" and $\theta^{t+1} = \theta^t - \lambda \nabla_{\theta^t} L(\theta^t)$ is used instead; L172, $\pi \rightarrow \rho$.

**R2: Third and fourth paragraph of Section 1, more clear motivation requires.** As suggested, we added: "It is hard to design a generic rule which can optimally match anchors/features with objects of various geometric layouts. The widely used hand-crafted assignment could fail when facing acentric, slender, and/or crowded objects. A learning-based approach requires to be explored to solve this problem in a systematic way."

**R2: Section 2, how the presented approach differs from existing methods (FoveaBox and MetaAnchor)?** FoveaBox used hand-crafted boxes to match features while MetaAnchor dynamically generated anchors from prior boxes. They remain using hand-crafted configuration to match objects with anchors/features, and thereby differs from our approach, essentially.

**R2: Section 3.1, what for was the IoU criterion used? What is similar/different to existing approaches. & What is the conclusion here (L110-113)? & Is this a limitation?** Section 3.1 is not our approach but a revisit of the baseline method which used the IoU criterion. The conclusion is that the matching matrix $C_{ij}$ in the baseline method is hand-crafted, not learned. In Section 3.2, we propose the learning-to-match approach to solve $C_{ij}$, and break the hand-crafted assignment limitation.

**R2: Why will it be close to the Mean when training is insufficient?** When training is insufficient, the "Mean" function assigns each anchor equal opportunity to be matched, which prevents the algorithm getting stuck into local minimum in early epochs.

**R2: The average performance across all categories & Why "couch" is a slender object? & Limitations** The average mAPs of square and slender objects are: 40.5%/40.3% and 22.7%/27.1% (baseline/ours), which show that FreeAnchor significantly improved the mAP on slender objects, while achieving comparable mAP on square objects. The average aspect ratio of couch is larger than 2.0. Limitation: FreeAnchor has no advantage on square objects as the IoU criterion can also find proper anchors for them.

**R2: Minor comments.** L92, "$R^4$ denotes {x, y, w, h}"; SmoothL1 was introduced in L103; "bg" indicates "background"; L121, "we require to guarantee..."; L156,"using the parameters..."; Bag construction were described in L118-120.

**R3: CenterNet is slightly better for large objects.** When using the same ResNet backbone, FreeAnchor outperforms state-of-the-art methods. CenterNet slightly outperforms FreeAnchor for large objects, as it uses a larger backbone (Hourglass) and a smaller input image size: 210.1M vs 96.9M parameters and $511 \times 511$ vs $1333 \times 800$ pixels (CenterNet vs Ours).

**R3: FreeAnchor and Anchor-Free.** "FreeAnchor" means that instances freely match anchors, without IoU restriction. To avoid the confusion with "Anchor-free", we revised "FreeAnchor" to "CatchAnchor": each instance can catch a proper anchor after training.

**R3: It would be better to see such a plug-and-play module could also help for the two-stage detectors.** During rebuttal, we have applied the learning-to-match module to a two-stage detector, FPN, and achieved significant (2.5%) performance gain.

[Meta-Review · NeurIPS 2019]

The paper presents a better loss function for anchor-based detection methods by matching anchors to GT boxes in a differentiable manner. Results are convincing. Three reviewers recommend acceptance after a convincing rebuttal. The final decision is to accept.